# Polydrug Use of Tobacco and Cannabis in a Cohort of Young People from Central Catalonia (2012–2020)

Eva Codinach-Danés [1,2,3], Núria Obradors-Rial [4,*], Helena González-Casals [4], Maria Rosa Cirera-Guàrdia [2], Queralt Miró Catalina [3], Josep Vidal-Alaball [1,3,5] and Albert Espelt [4,6]

1  Health Promotion in Rural Areas Research Group, Gerència Territorial de la Catalunya Central, Catalan Health Institute, C/Pica d'Estats 13-15, 08272 Sant Fruitós de Bages, Spain; ecodinach.cc.ics@gencat.cat (E.C.-D.); jvidal.cc.ics@gencat.cat (J.V.-A.)
2  Centre d'Atenció Primària Sant Quirze de Besora, Gerència Territorial de la Catalunya Central, Institut Català de la Salut, Pg del Ter 21, 08580 Sant Quirze de Besora, Spain; mrcirera.cc.ics@gencat.cat
3  Unitat de Suport a la Recerca de la Catalunya Central, Institut Universitari d'Investigació en Atenció Primària Jordi Gol, C/Pica d'Estats 13-15, 08272 Sant Fruitós de Bages, Spain; qmiro.cc.ics@gencat.cat
4  Department of Epidemiology and Methodology of Social and Health Sciences, Faculty of Health Sciences of Manresa, Universitat de Vic—Universitat Central de Catalunya (UVic-UCC), Av. Universitària 4-6, 08242 Manresa, Spain; hgonzalez@umanresa.cat (H.G.-C.); aespelt@umanresa.cat (A.E.)
5  Faculty of Medicine, University of Vic-Central University of Catalonia, 08500 Vic, Barcelona, Spain
6  Centro de Investigación Biomédica en Red de Epidemiología y Salud Pública (CIBERESP), C/Monforte de Lemos 3 Pabellón 11, 28029 Madrid, Spain
*  Correspondence: nobradors@umanresa.cat; Tel.: +34-93-8774179

**Abstract:** The aim of this study was to estimate the prevalence of tobacco and cannabis use, and their relationship, in a cohort of adolescents from Central Catalonia in the period 2012–2020. The study had a prospective longitudinal design with 828 students in the 4th year of Compulsory Secondary Education. In 2012, 828 adolescents answered a health behaviour survey; in 2016, 342; and in 2020, 265. The dependent variables were: exclusive last month tobacco use; exclusive last month cannabis use; polydrug use of tobacco and cannabis; and no use of tobacco and cannabis in the last month. Independent variables were sex and follow-up year. For the analysis, prevalences were used with a significance level of 0.05. Polydrug use went from 11.5% (95% CI: 7.4–17.6) among girls and 8.2% (95% CI: 4.3–15.2) among boys in 2012, to 7.0% (95% CI: 3.9–12.3) among girls and 13.7% (95% CI: 8.4–21.7) among boys in 2020. With regards to those girls who used polydrug in the last month in 2012, 33.3% (14.8–58.9) continued using it in 2016, and 22.2% (7.9–48.6) continued in 2020; among boys, 33.3% (14.8–58.9) of those polydrug users in 2012 continued using it in 2016, and 44.4% (14.5–79.0) continued in 2020. There is association between tobacco and cannabis use, and there is an important percentage of young people who do polydrug consumption at very early ages.

**Keywords:** tobacco; cannabis; cohort study; young adults

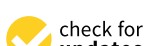



## 1. Introduction

Tobacco and cannabis use is considered a public health problem [1]. Both are commonly used among the general population, and especially among young people [2]. Specifically, tobacco is the most consumed psychoactive substance among students between 14 and 18 years of age in Secondary Education in Spain. In 2020, 26.7% of students aged 14–18 years self-reported as having smoked tobacco in the last 30 days [3]. Tobacco consumption is more frequent in girls, in particular for daily consumption (10.3% among girls compared to 9.4% in boys). Cannabis is the most prevalent illegal psychoactive substance used among students aged 14–18. Within the last 30 days, 19.3% [3] of students confirmed their cannabis use. Cannabis use is more widespread among boys. In this group, 15.8% of 14-year-old students have already used cannabis, a proportion that increases progressively with age, resulting in more than the half of 18-years-old having already consumed cannabis

(51.2%) [4]. Often, we also find consumption of the two substances [4], which we refer to as polydrug use. This is understood as being the use of several substances during the same period of time (throughout the previous month), regardless of whether they have been used simultaneously, or alternately [5]. The use of cannabis and tobacco simultaneously over time means adding to the risks of each substance, in particular those produced by mixing different drugs [6]. Previous evidence shows that simultaneous use of both substances is associated with a higher likelihood of suffering from psychosocial problems [7], it is also associated with being much more likely to suffer from personality disorders and antisocial behaviors [8]. Epidemiological data on tobacco and cannabis use in Spain shows that there is a strong relationship between the use of these substances [2,7,9,10]. The likelihood of using tobacco, cannabis or even alcohol increases when one of the other two are also used [5]. In this regard, several studies show that young people who smoke tobacco are more likely to use cannabis than those who do not smoke [11–13]. This is due to the fact that tobacco use is one of the most common forms of consumption, which starts at an early age and is considered a gateway to other substances [14–16]. Typically, it was assumed that the onset of cannabis use followed the onset of tobacco use [14–16]. Currently, there are studies that show that the onset of the two consumptions can be simultaneous [14–16]. It can also be the other way around, starting with cannabis use and then continuing with tobacco use, i.e., a "reverse" gateway from cannabis to tobacco. Therefore, cannabis users are also at a greater risk of initiating tobacco use [17] and nicotine dependence [17,18].

There are many studies that refer to the prevalence of tobacco and cannabis independently, but given that most risk and prevention factors are applicable to both substances, it is important to identify common prevention strategies for both. A deeper knowledge of the factors related to polydrug use, and its implications on both sexes, is necessary in order to adapt preventive strategies for new scenarios. The aim of this study is to estimate the prevalence of tobacco and cannabis use, and the relationship between these two substances, in a cohort of young people from Central Catalonia over the period 2012–2020, according to sex and year of follow-up.

## 2. Materials and Methods

### 2.1. Study Design

This was a prospective longitudinal study based on a sample of 828 students in the 4th year of Compulsory Secondary Education (ESO) (15–16 years) in Central Catalonia. Pupils in the academic year 2011–2012, and across 26 schools (in the counties of Bages, Moianès, Anoia, Osona, Berguedà and Solsonès), completed a computerised and self-administered questionnaire on health behaviour.

Participants that consented to follow-up were contacted again in 2016 (19–20 years), of which 342 participants (41.3%) took part. The final follow-up was conducted in 2020 (23–24 years), of which 265 students took part (32%) (Figure 1). Participants were contacted by email or telephone. They were sent the study information and gave informed consent by e-mail. The e-mail included a link to the questionnaire. If the participant didn't accept the informed consent, they could not begin the questionnaire. We would also like to point out that, in 2016 (19–20 years old), the students who agreed to participate again were asked to use their data from 2011–2012 and agree to be followed in future studies. The data from other students who participated on 2011–2012, but who did not agree to participate, is not shown in the present study. In 2016 a specific informed consent was required. The questionnaire used in 2012 was created specifically for this study, using previously validated questionnaires on various behaviours and lifestyles. This had a response time of about 40 min. The questionnaire was administered by staff from outside the school, who provided instructions for completing it and answered possible questions. Subsequently, in 2016 and 2020, a scaled-back version of the same questionnaire was used, adapted on-line through the LimeSurvey tool (Available online: https://www.limesurvey.org/) (accessed on 1 July 2022), and self-administered.

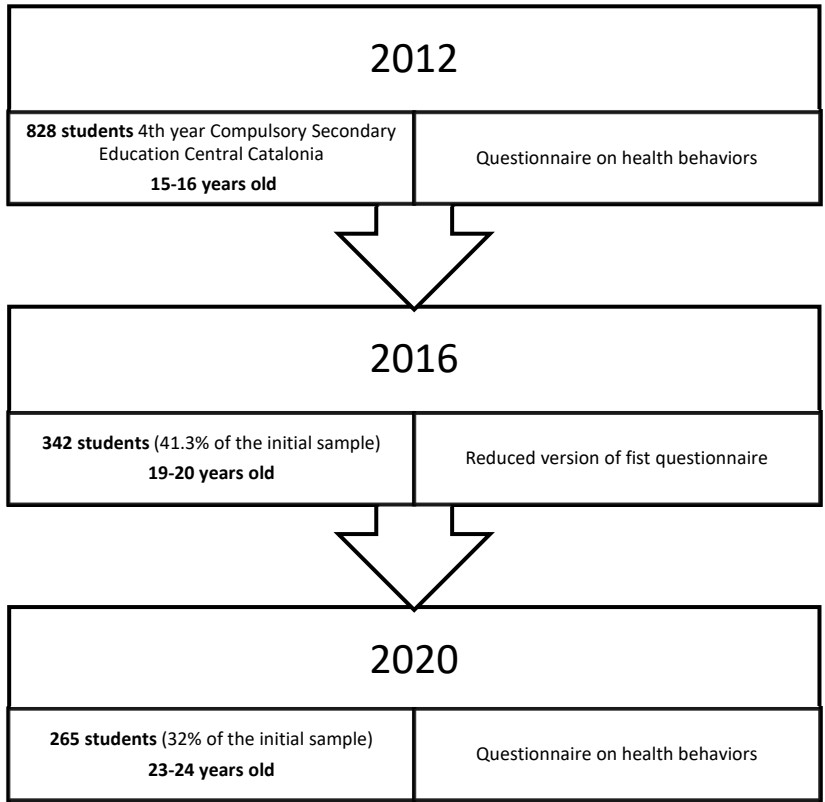

**Figure 1.** General outline of the study.

*2.2. Study Variables*

There are 4 dependent variables: exclusive last month tobacco use; exclusive last month cannabis use; polydrug use of tobacco and cannabis; and no use of tobacco and cannabis in the last month.

The 'exclusive last month tobacco use' variable was constructed using the following question: "How much have you smoked in the last 30 days?" Four response options were offered: every day; once or more times a week; less than once a week; I have not smoked in the last 30 days. Those who answered that they smoked every day, once or more times a week, or less than once a week, and those who answered to the question: "Have you ever used hashish or marijuana (joints, cannabis, weed, chocolate, hash, reefer or joints)?" and answered: never used, or sometimes in your life, or in the last 12 months, were classified as exclusive last month tobacco users. The variable 'exclusive last month cannabis use' was constructed using the following question: "Have you ever used hashish or marijuana (joints, cannabis, weed, chocolate, hash, reefer or joints)?" Four possible answers were offered: never used; used sometimes in your life; used in the last 12 months; or used in the last 30 days. Those who responded that they had used hashish or marijuana in the last 30 days, and those who answered that they had smoked in the last 30 days were classified as having used cannabis in the last month. The 'polydrug use of both substances' variable was constructed from last month tobacco and cannabis uses. The 'no use of tobacco and cannabis in the last month' variable was constructed with those who answered that they had not smoked tobacco in the last 30 days and answered: never used, or sometimes in your life, or in the last 12 months, regarding the use of cannabis.

As independent variables, sex (boy, girl) and year of follow-up (2012, 2016, 2020) were taken into account.

*2.3. Statistical Analysis*

The prevalence of each of the dependent variables was compared with the independent variables using relative frequencies and percentages, with their respective 95% confidence intervals. A significance level of 0.05 (two tailed) was set. Analyses were performed using STATA 16 software.

## 3. Results

Table 1 shows the prevalence of exclusive last month tobacco use, exclusive last month cannabis use, polydrug use of tobacco and cannabis, and no use of tobacco and cannabis in the last month. In 2012, 22.4% (95% CI: 16.5–29.7) of girls and 15.6% (95% CI: 9.9–23.8) of boys used tobacco in the last month, with this value increased in 2020 among girls to 26.9% (95% CI: 20.5–34.5) and among boys to 20.2 (95% CI: 13.6–28.9).

**Table 1.** Prevalence of exclusive use according to substance consumed (last month tobacco use, last month cannabis use, polydrug use of tobacco and cannabis and no use of tobacco and cannabis in the last month), sex and year. Cohort 2012–2020.

| | Girls | Boys | Total |
|---|---|---|---|
| | % [95% CI] *n* = 156 | % [95% CI] *n* = 109 | % [95% CI] *n* = 265 |
| Last month tobacco use [a] | | | |
| 2012 | 22.4 (16.6–29.7) | 15.6 (9.9–23.8) | 19.6 (15.2–24.9) |
| 2016 | 30.8 (24.0–38.5) | 22.9 (15.9–31.8) | 27.5 (22.4–33.3) |
| 2020 | 26.9 (20.5–34.5) | 20.2 (13.6–28.9) | 24.1 (19.3–29.7) |
| Last month cannabis use [b] | | | |
| 2012 | 0.6 (0.08–4.4) | 2.7 (0.8–8.3) | 1.5 (0.5-3.9) |
| 2016 | 0 | 2.7 (0.8–8.3) | 1.1 (0.3–3.5) |
| 2020 | 1.9 (0.6–5.8) | 2.7 (0.8–8.3) | 2.3 (1.0–5.0) |
| Polydrug use of tobacco and cannabis [c] | | | |
| 2012 | 11.5 (7.4–17.6) | 8.2 (4.3–15.2) | 10.2 (7.1–14.5) |
| 2016 | 8.3 (4.9–13.9) | 15.6 (9.9–23.8) | 11.3 (8.0–15.7) |
| 2020 | 7.0 (3.9–12.3) | 13.7 (8.4–21.7) | 9.8 (6.7–14.0) |
| No use of tobacco and cannabis in the last month [d] | | | |
| 2012 | 65.4 (57.5–72.5) | 73.4 (64.2–80.9) | 68.7 62.9–74.0) |
| 2016 | 60.9 53.0–68.3) | 58.7 (49.2–67.6) | 60.0 (53.9–65.7) |
| 2020 | 64.1 (56.2–71.3) | 63.3 (53.8–71.9) | 63.8 (6.7–57.8–69.4) |

[a] Last month tobacco use: smoking every day; once or more times a week; less than once a week; and never used cannabis in the last 12 months. [b] Last month cannabis use: use of cannabis in the last 30 days; and not smoked in the last 30 days. [c] Polydrug use of tobacco and cannabis: use of tobacco and cannabis in the last month. [d] No use of tobacco and cannabis in the last month: not smoked tobacco in the last 30 days; and never used cannabis in the last 12 months.

Cannabis use increased from 0.6% (95% CI: 0.08–4.4) among girls who had used it exclusively in the last month in 2012 to 1.9% (95% CI: 0.6–5.8); consumption remained stable in the case of boys from the beginning, 2.7 (95% CI: 0.8–8.3) (2012), to the end of follow-up, 2.7 (95% CI: 0.8–8.3) (2020).

With regards to polydrug use (last month tobacco and last month cannabis use), it went from 11.5% (95% CI: 7.4–17.6) among girls and 8.2% (95% CI: 4.3–15.2) among boys in 2012, to 7.0% (95% CI: 3.9–12.3) among girls and 13.7% (95% CI: 8.4–21.7) among boys in 2020.

In the case of no use of tobacco and cannabis in the last month it went from 65.4% (95% CI: 57.5–72.5) among girls and 73.4% (95% CI: 64.2–80.9) among boys in 2012, to 64.1% (95% CI: 56.2–71.3) among girls and 63.3% (95% CI: 53.8–71.9) among boys, at the end of follow-up (2020) (Table 1).

Table 2 shows the prevalence of polydrug use of tobacco and cannabis in the last month in 2016 and 2020 by frequency of: exclusive last month tobacco use in the last month; exclusive last month cannabis use; polydrug use of tobacco and cannabis in the last month; and no use of tobacco and cannabis in the last month, compared to figures from 2012. We observed that 14.3% (5.9–30.8) of girls using tobacco in the last month in 2012, used polydrug in 2016, and 2.8% (40.3–18.8) used polydrug in 2020. Of those who consumed cannabis in the last month in 2012, 0% used polydrug in 2016 and 2020. With regards to girls' polydrug use in the last month in 2012, 33.3% (14.8–58.9) and 22.2% (7.9–48.6) continued to consume polydrug in 2016 and 2020. Of those who did not use tobacco and cannabis in the last month in 2012, 1.9% (0.4–7.7) and 5.8% (2.6–12.6) used polydrug in 2016 and 2020, respectively. With regards to the boys, of those who used tobacco in the last month in 2012, 17.6% (5.3–46.2) and 5.9% (0.6–35.2) used polydrug in 2016 and 2020, respectively. Of those who consumed cannabis in the last month in 2012, 33.3% (0.2–9.9) and 66.7% (10.2–99.7) used polydrug in 2016 and 2020, respectively. With regards to those boys who used polydrug in the last month in 2012, 33.3% (14.8–58.9) and 44.4% (14.5–79.0) continued to consume polydrug in 2016 and 2020. Of those who did not use tobacco and cannabis in the last month in 2012, 12.5% (6.7–21.8) and 10.0% (5.0–18.9) did polydrug use in 2016 and 2020, respectively (Table 2).

**Table 2.** Prevalence of polydrug use of tobacco and cannabis in the last month in 2016 and 2020 by frequency of: exclusive last month tobacco use; exclusive last month cannabis use; polydrug use of tobacco and cannabis in the last month; and no use of tobacco and cannabis in the last month, compared to figures from 2012. Cohort 2012–2020.

| | Girls | | Boys | | Total | |
|---|---|---|---|---|---|---|
| | 2016 | 2020 | 2016 | 2020 | 2016 | 2020 |
| Last month tobacco use 2012 [a] | % (95% CI) | % (95% CI) | % (95% CI) | % (95% CI) | % (95% CI) | % (95% CI) |
| Polydrug use of tobacco and cannabis | 14.3 (5.9–30.8) | 2.8 (0.3–18.8) | 17.6 (5.3–45.2) | 5.9 (0.6–35.2) | 15.4 (7.7–28.2) | 3.8 (0.9–14.5) |
| Last month cannabis use [b] | % [95% CI] | % [95% CI] | % [95% CI] | % [95% CI] | % [95% CI] | % [95% CI] |
| Polydrug use of tobacco and cannabis | 0 | 0 | 33.3 (0.2–9.9) | 66.7 (10.2–99.7) | 25.0 (0.8–92.9) | 50.0 (39.8–96.0) |
| Polydrug use of tobacco and cannabis in 2012 [c] | % [95% CI] | % [95% CI] | % [95% CI] | % [95% CI] | % [95% CI] | % [95% CI] |
| Polydrug use of tobacco and cannabis | 33.3 (14.8–58.9) | 22.2 (7.9–48.6) | 33.3 (8.9–71.8) | 44.4 (14.5–79.0) | 33.3 (17.8–53.6) | 29.6 (15.0–50-0) |
| No use of tobacco and cannabis in the last month of 2012 [d] | % [95% CI] | % [95% CI] | % [95% CI] | % [95% CI] | % [95% CI] | % [95% CI] |
| Polydrug use of tobacco and cannabis | 1.9 (0.4–7.6) | 5.8 (2.6–12.6) | 12.5 (6.7–21.8) | 10.0 (5.0–18.9) | 6.6 (3.8–11.3) | 7.7 (4.6–12.6) |

[a] Last month tobacco use in 2012: smoking every day; once or more times a week; less than once a week; and never used cannabis in the last 12 months. [b] Last month cannabis use in 2012: use of cannabis in the last 30 days; and not smoked in the last 30 days. [c] Polydrug use of tobacco and cannabis: use of tobacco and cannabis in the last month in 2012. [d] No use of tobacco and cannabis in the last month in 2012: not smoked tobacco in the last 30 days; and never used cannabis in the last 12 months.

## 4. Discussion

The main results of this study show that the substance that increases the most over the years is last month tobacco use. This consumption was higher among girls. The most pronounced increase was found mainly between the ages of 15–16 and 19–20, in both sexes. Exclusive cannabis use remains stable over the years. In boys, with respect to polydrug

use, we found an increase between 15–16 and 19–20 years, and after that, it remains stable. In girls, it decreases from 15–16 to 19–20 years, and then it remains stable.

In line with studies carried out in similar environments, in our study we also found that tobacco consumption is more frequent in the female group; it seems that they use it to appear more attractive and it provides them with a good image [19], although the greatest increase in consumption with age is found in the male group [3].

Regarding cannabis use and polydrug use, we found that in the group of girls there seems to be a decrease in use, although this is not significant, from 15–16 to 19–20 years, which can be explained by the phenomenon of experimentation [20] that occurs at school age, but is not observed in adults. This could also be because girls have a greater perception of the risks associated with drug use [21] and, consequently, greater concern about the negative impact caused by drugs on their own health [22]. It could also be due to them taking on the roles and responsibilities associated with being more adult [23]. In the male group, consumption of polydrug increases from 15–16 to 19–20 years, but cannabis use remains stable during this period. Other studies have found that male adolescents reported various motivations for consuming cannabis, including that it makes it easier for them to achieve an optimal emotional state, and helps them to flirt or socialize [24].

The results of the study show there is a percentage of boys who are initiated into the use of smoking drugs by reverse entry i.e., those who smoked cannabis and had not smoked tobacco, 33.3% (95% CI: 0.2–9.9) and 66.7% (95% CI: 10.2–99.7) in 2012, but used polydrug in 2016 and 2020, respectively, reinforcing the theory that tobacco use no longer always precedes cannabis use. This reverse onset of use may also be explained by the fact that they are ex-smokers of tobacco, and at the time of the survey only used cannabis. These results are in line with other studies that show that the age of initiation of both drugs, or even initiation with only cannabis, is becoming more and more common as a consequence of the increasing availability of this substance [14–17].

The main limitation of this study is the significant sample loss over the years, which is very common in longitudinal studies like this one, especially when involving young people. Also, although we have a longitudinal study and we could have observed individual changes in the study, only changes in prevalence have been analysed. Another current limitation is that the survey is self-reported and social desirability, or difficulty in remembering one's own behaviour, may have an impact on responses; however, there is evidence that the use of self-administered questionnaires is a viable method for measuring substance use variables among adolescents [25]. Also, it needs to be considered that the dependent variable of exclusive tobacco use does not incorporate the number of cigarettes smoked. Finally, we haven't collected data on other forms of tobacco and cannabis consumption, such as the electronic cigarettes and different methods of cannabis administration, such as vaping, cannabis oil or wax; however, these forms of consumption are less frequent when talking about polydrug use of tobacco and cannabis.

## 5. Conclusions

Among the main conclusions, we observe on the one hand that there is a relevant number of young people using polydrug with tobacco and cannabis. On the other hand, there is an important percentage of young people who do polydrug consumption at very early ages, and we observe that this acts as a factor associated to their future polydrug consumption, so they continue to do so at older ages. Considering the change in the context of tobacco and cannabis use, and its strong association, it is important to take into consideration the need to start implementing training programs at an early age directly aimed at the prevention of polydrug use of tobacco and cannabis. For all these reasons, it is very important to improve educational public health policies focused on reducing the initiation, and the harm caused by the use of these two substances [5].

**Author Contributions:** Conceptualization, E.C.-D., N.O.-R. and A.E.; methodology, E.C.-D., N.O.-R. and A.E.; software, A.E. and H.G.-C.; validation, E.C.-D., N.O.-R. and A.E.; formal analysis, Q.M.C. and J.V.-A.; investigation: E.C.-D., N.O-R. and A.E., writing—original draft preparation, E.C.-D., N.O.-R. and A.E.; writing—review and editing, E.C.-D., N.O.-R., Q.M.C., M.R.C.-G., J.V.-A. and A.E.; supervision, E.C.-D., N.O.-R. and A.E. All authors have read and agreed to the published version of the manuscript.

**Funding:** This research received no external funding. In 2018 the author received the ICS Scholarship from the Territorial Management of Central Catalonia for Research Training and Doctoral Completion in Primary Care. The APC was funded by Catalan Health Institute.

**Institutional Review Board Statement:** The study was conducted according to the guidelines of the Declaration of Helsinki [26] and approved by the Research Ethics Committee: Fundació la Unió Catalana d'Hospitals (codes: 16/31, approved on 4 April 2016).

**Informed Consent Statement:** In 2016, an email was sent informing each participant of the continuation of the study initiated in the 2011–2012 academic year. The students were contacted and informed that the current study was a follow-up of the data collected in 2011–2012, and the intention to contact them again in the future (2019–2020) with the objective being to describe changes across the years. Only data from students who agreed to participate in the study was included in the current study. An informed consent was obtained from all of them. The e-mail also explained that participation was voluntary, and that they could drop out whenever the participant wished without having to give any explanation. The data collected by the study were treated with complete confidentiality, were identified by a numerical code, and only the study investigators could link these data to the participant.

**Data Availability Statement:** The data presented in this study are available upon reasonable request to the corresponding author. The data are not publicly available due to confidentiality reasons.

**Acknowledgments:** The authors would like to thank the young people who gave their consent to continue participating in the study, and who have made it easier for the researchers to carry out the new study. This article is part of the thesis that Eva Codinach-Danés is carrying out within the doctoral programme in Comprehensive Care and Health Services of the Univeristat de Vic-Universitat Central de Catalunya.

**Conflicts of Interest:** The authors declare no conflict of interest.

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
