# Peer review of "Polydrug Use of Tobacco and Cannabis in a Cohort of Young People from Central Catalonia (2012–2020)"

_adolescents, doi:10.3390/adolescents2030027_

Round 1

Reviewer 1 Report

Congratulations for this informative and well written article. Conclusions can give provide data for tobacco control policymaking.

Please provide in the conclusions section, and analysis of the bases that can be produced by the lost of population during the followup process.

Reviewer 2 Report

Thank you for the opportunity to review this interesting manuscript. I agree with the authors that the dual use of tobacco and cannabis it is an important topic. I also like the introduction section of this manuscript. However in my opinion, methods, results and discussion needs to be improved. See bellow in detail my comments.

Introduction

1.     If the prevalence of smoking in the last month is 26.4% for students aged 14-18, how is possible that smoking among girls is only 10.3% and among boys only 9.4%.  Is there a different smoking definition? Or is there a different age group? The authors needs to clarify this sentence.  Also the authors said that tobacco use is more frequent among girls 10.3% than 9.4%.   Is this difference statistical significant? If not the authors should change the language because 10.3% and 9.4% is similar.

Page 1 “In 2020, 26.7% of students aged 14-18 years self-reported to having smoked tobacco in the last 30 days [3]. Tobacco consumption is more frequent among girls (10.3%, compared to 9.4% among boys)”

Methods

2.     This is prospective longitudinal cohort. This studies are ideal to evaluate transitions between products. However, the authors decided to evaluate changes in prevalence estimates which is a more approach that it is more appropriate for cross-sectional studies.  I am not suggesting that the authors should change their methodology, but this limitation of using prevalence estimates needs to be acknowledged in the discussion section.

3.     In the context of dual use of substance use behaviors. I believe that a more clean definition would be to create a variable of four categories of current use in the last month (no use of tobacco and cannabis, exclusive tobacco use, exclusive cannabis use, and dual use of tobacco and cannabis. I strongly suggest to make the analysis in this way, would be easy for the readers to follow.

Results

4.     The description of the results is unclear. I suggest to incorporate my suggestions for the methods, this will help to improve the descriptions of the results.

5.     It is interesting to observe that the use of tobacco and cannabis is not increasing over time. In fact, the highest prevalence is when the adolescents were 15-16 years old. The authors recognize that this result potentially can be explained because the adolescents of 15-16 years are in the process of experimentation. Therefore, it is possible that the authors are using a sample with many infrequent users, and this could explain that the prevalence of tobacco and cannabis is higher or similar for 15-16 years adolescents than 23-24 years adults. I suggest to the authors to do a sensitivity analysis by excluding experimental users. Ideally, the authors could create a variable that can evaluate established use. For example, in tobacco community the definition of established use included those who have smoked more than 100 cigarettes in their life and smoked in the last month or those who reported that they have smoked regularly. Is there established use questions in your study? If yes, please follow my suggestion. If there is no questions of established use, please include this as a limitation.

6.     Results and Discussion should be reframe after the implementation of my suggestions in the methods section

Round 2

Reviewer 2 Report

The authors responded appropriately to reviewers' comments. The improvement in data understandability with the changes in the definition of the dependent variable is excellent. My only comment is to review table one. There seems to be a typo (2012, last month, cannabis use was 12.2% of the total, but only 0.6% for girls and 2.7% for boys). Please check these numbers.